# Real-time observation of Cooper pair splitting showing strong non-local correlations

Antti Ranni [1]✉, Fredrik Brange [2], Elsa T. Mannila [2], Christian Flindt [2] & Ville F. Maisi [1]✉

Controlled generation and detection of quantum entanglement between spatially separated particles constitute an essential prerequisite both for testing the foundations of quantum mechanics and for realizing future quantum technologies. Splitting of Cooper pairs from a superconductor provides entangled electrons at separate locations. However, experimentally accessing the individual split Cooper pairs constitutes a major unresolved issue as they mix together with electrons from competing processes. Here, we overcome this challenge with the first real-time observation of the splitting of individual Cooper pairs, enabling direct access to the time-resolved statistics of Cooper pair splitting. We determine the correlation statistics arising from two-electron processes and find a pronounced peak that is two orders of magnitude larger than the background. Our experiment thereby allows to unambiguously pinpoint and select split Cooper pairs with 99% fidelity. These results open up an avenue for performing experiments that tap into the spin-entanglement of split Cooper pairs.

[1] NanoLund and Solid State Physics, Lund University, Box 118, 22100 Lund, Sweden. [2] Department of Applied Physics, Aalto University, 00076 Aalto, Finland. ✉email: antti.ranni@ftf.lth.se; ville.maisi@ftf.lth.se

Cooper pair splitters are promising solid-state devices for generating nonlocal entanglement of electronic spins[1,2]. The basic operating principle is based on the tunneling of spin-entangled electrons from a superconductor into spatially separated normal-state structures, whereby entanglement between different physical locations is obtained. The controlled generation of entangled particles is not only of fundamental interest to test the foundations of quantum mechanics[3]. It is also a critical prerequisite for future quantum technologies[4,5], such as quantum information processors and other quantum devices, which will be "fueled" by entanglement.

Until now, the splitting of Cooper pairs has been indirectly inferred from measurements of the currents in the outputs of a Cooper pair splitter[6–26] or their low-frequency cross-correlations[12,15]. These approaches rely on measuring the average currents, or small fluctuations around them, due to a large number of splitting events. The currents consist of contributions from Cooper pair splitting as well as other competing transport processes, which not only makes these experiments highly challenging, but also hinders direct access to the correlated electron pairs. In particular, a considerable fraction of the electrical signal is due to unwanted processes, and those electrons that actually are entangled have already passed through the device and are lost, once the currents have been measured.

Here, we take a fundamentally different route and use charge detectors to observe the splitting of individual Cooper pairs in real time as it happens. Unlike earlier experiments, we use isolated islands that are not connected to external drain electrodes. Hence, once a Cooper pair is split, the two correlated electrons remain stored on the islands and can be detected in real time.

## Results

**Device architecture.** Figure 1a shows a scanning electron microscope image of one of our Cooper pair splitters made of a superconducting aluminium electrode coupled to two normal-state copper islands. The grounded superconducting electrode and the islands are connected via insulating aluminium oxide layers acting as tunnel junctions. The superconductor functions as a source of Cooper pairs, which are split into two separate electrons that tunnel into the islands, one in each. We detect the number of electrons on each island using single-electron transistors whose conductance depends sensitively on the charge state of the islands[27–29]. Each detector is biased by a voltage $V_{D\alpha}$ ($\alpha = L$ or $R$) and the currents $I_{D\alpha}$ through them are measured in the grounded contacts. The detectors are tuned to charge-sensitive operation points by the gate voltages $V_{DG\alpha}$. We can thus monitor the tunneling of electrons in and out of the islands, including simultaneous processes[30,31], such as Cooper pair splitting, where one electron tunnels into each of the two islands at the same time. The electronic population of each island is feedback-controlled by the gate voltages $V_{G\alpha}$, which are adjusted to maintain the two lowest-lying charge states at equal

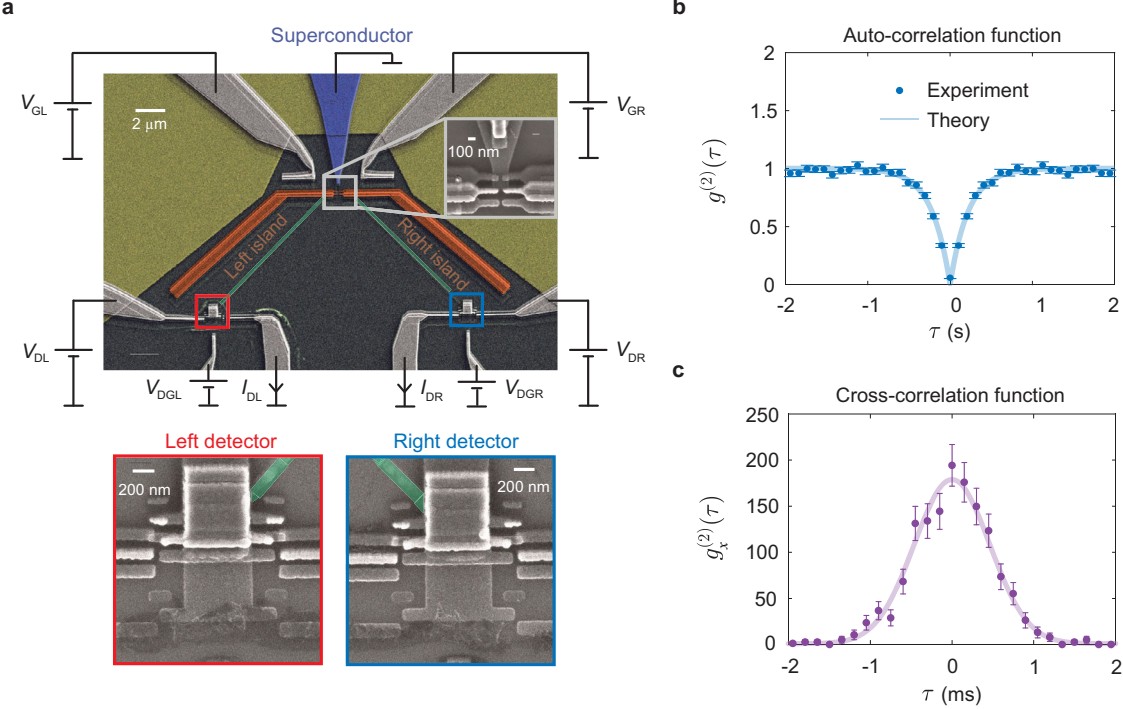

**Fig. 1 Cooper pair splitter and correlation measurements. a** Scanning electron micrograph of the grounded superconducting reservoir (colored blue) coupled via tunnel junctions to two normal-state metallic islands (in orange). The voltages $V_{GL}$ and $V_{GR}$ are connected to gate electrodes and tuned to control the electronic populations on the islands. Each island is capacitively coupled via a metal strip (in green) to a single-electron transistor, which serves as a real-time charge detector to read out the number of electrons on each island from the measured signals $I_{DL}$ and $I_{DR}$. The detectors are biased by the voltages $V_{D\alpha}$ ($\alpha = L$ or $R$) and gated by the voltages $V_{DG\alpha}$. The inset in the upper right corner shows the tunnel junctions that connect the superconductor to the two metallic islands. **b** Measurements of the auto-correlation function $g^{(2)}$ for single-electron tunneling into the right island with a time delay $\tau$ between tunneling events. The correlation function is suppressed at short times due to the strong Coulomb interactions on the island, which lead to anti-bunching. **c** Cross-correlation function $g_x^{(2)}$ for transitions between the superconductor and the left island at the time $t = 0$, followed by transitions between the superconductor and the right island at the time $t = \tau$. The cross-correlations are strongly enhanced for short times due to crossed two-electron processes such as Cooper pair splitting and elastic cotunneling between the islands via the superconductor. The error bars in (**b**, **c**) are given by the standard deviations. Details of the device fabrication and the measurements are provided in "Methods".

energies[32,33]. Hence, there is no energy cost for tunneling processes that change the electron number on each island by one.

**Correlation measurements.** Figure 1b and c shows correlation measurements using our charge detectors. The $g^{(2)}$-function describes correlations between tunneling processes with a time delay $\tau$ between them and is, for example, used extensively in quantum optics to characterize light sources[34]. A value of $g^{(2)}(\tau) = 1$ at all times implies that the particles are uncorrelated. On the other hand, a $g^{(2)}$-function that peaks at short times indicates that the particles tend to bunch as for instance for thermal light, which has $g^{(2)}(0) = 2$ at $\tau = 0$. By contrast, a coherent single-photon source is characterized by a $g^{(2)}$-function with a dip at short times, indicating that the particles are anti-bunched, ideally with $g^{(2)}(0) = 0$.

Here, we measure the auto-correlations for tunneling events from the superconductor to the right island as well as the cross-correlations for tunneling between the superconductor and each of the two islands. The auto-correlations in Fig. 1b are fully suppressed at short times. The suppression arises from the strong Coulomb interactions, which prevent more than one electron at a time to tunnel into the right island, leading to anti-bunching of the tunneling events. A theoretical analysis captures the experimental findings in Fig. 1b by the expression

$$g^{(2)}(\tau) = 1 - e^{-\gamma|\tau|}, \tag{1}$$

where $\gamma = 4.5\,\text{s}^{-1}$ is the inverse correlation time, which may be determined from the tunneling rates (see Supplementary Note 2 for details). Equation (1) is similar to the auto-correlation function of a simple two-level system, however, in our case, it follows from an elaborate model of the Cooper pair splitter, which involves several charge states of each island, and the inverse correlation time is a complicated function of all tunneling rates.

Turning next to the cross-correlations in Fig. 1c, a completely different picture emerges. We now consider the conditional probability of observing a tunneling event between the superconductor and the right island at the time delay $\tau$ after a tunneling event between the superconductor and the left island has occurred. In this case, we observe a large peak at short times which is two orders of magnitude larger than the uncorrelated background of $g_x^{(2)}(\tau) = 1$. These correlations are a direct manifestation of nearly instantaneous two-electron processes

involving both islands. The two-electron processes lead to correlated single-electron events in the two islands occurring on a microsecond timescale, which is much faster than the correlation time of the single-electron processes for each island as seen in Fig. 1b. Theoretically, we can describe the cross-correlations as

$$g_x^{(2)}(\tau) = 1 + \alpha_2 \frac{e^{-\frac{1}{2}(\tau/\sigma_D)^2}}{\sqrt{2\pi}\sigma_D}, \tag{2}$$

where $\sigma_D = 460\,\mu\text{s}$ is the broadening due to timing jitter of the detectors, and $\alpha_2 = 210\,\text{ms}$ is the time-integrated contribution from two-electron processes. Figure 1c shows that this expression agrees well with the experiment, explaining the strong nonlocal correlations. Figure 1 also illustrates how the cross-correlations, unlike the auto-correlations, can be measured on a timescale, which is shorter than the rise time of each detector of about 4 ms: The cross-correlations concern tunneling events in different islands and are not limited by the dead time of the detectors (see Supplementary Note 1).

**Cooper pair splitting in real time.** We now address the identification of the individual tunneling events. Typical time traces of the electrical currents in the two single-electron transistors are shown in Fig. 2a, illustrating how we can detect the tunneling events in each island. At the time marked by ①, the current in the left detector (red curve) switches suddenly from 60 to 80 pA as an electron tunnels out of the left island as depicted in panel b. At the time marked by ②, the current in the left detector switches again as an electron tunnels into the left island. However, this time, the current in the right detector (blue curve) also changes as an electron tunnels into the right island. As we discuss below, statistical arguments allow us to conclude with the near-unity probability that these simultaneous tunneling events occur due to the splitting of a Cooper pair as indicated in panel b, rather than being two uncorrelated single-electron processes. Continuing further in time, the point marked by ③ again corresponds to the tunneling of a single electron as in ①, except that now an electron tunnels out of the right island. Finally, at the time marked by ④, the detector currents switch in opposite directions due to elastic cotunneling in which an electron tunnels from the left to the right island via the superconductor.

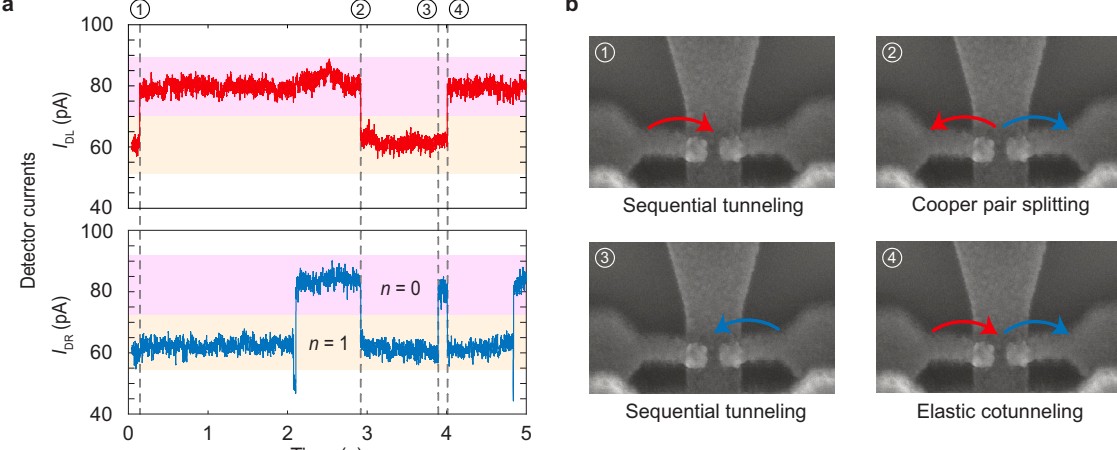

**Fig. 2 Real-time observation of Cooper pair splitting. a** Typical time traces of the currents $I_{DL}$ and $I_{DR}$ in the left and the right single-electron transistors correspondingly, which switch between two distinct levels corresponding to having $n = 0$ or 1 (excess) electrons on each island. **b** The points marked with ① and ③ in the left panel correspond to processes, where a single electron tunnels between the superconductor and one of the islands. The point marked with ② is a Cooper pair splitting event, where two electrons simultaneously tunnel from the superconductor into the two islands, one in each, while the point marked with ④ is an elastic cotunneling event, where an electron is transferred between the two islands via the superconductor.

We now return to the alleged Cooper pair splitting at the time marked by ②. Strictly speaking, the two simultaneous tunneling events could in principle be completely uncorrelated. However, to rule out that scenario, we again consider our cross-correlation measurements in Fig. 1c. Uncorrelated single-electron processes give rise to a flat background with $g_x^{(2)}(\tau) = 1$, while correlated two-electron processes, such as Cooper pair splitting, lead to the pronounced peak at short delay times, and they essentially all take place within a time window of width $6\sigma_D \simeq 3$ ms. Within this time window, the fraction of uncorrelated single-electron processes is $6\sigma_D/(\alpha_2 + 6\sigma_D)$, while the fraction of correlated two-electron processes is $\alpha_2/(\alpha_2 + 6\sigma_D)$. Hence, with fidelity $\mathcal{F} = \alpha_2/(\alpha_2 + 6\sigma_D) \simeq 99\%$ we can say that the point marked by ② represents the splitting of a Cooper pair.

**Waiting-time distributions**. Based on the identification of the involved tunneling processes, we can determine all relevant tunneling rates in our experiment. Furthermore, additional information about the tunneling processes can be obtained from measurements of the electron waiting times[35–37]. Figure 3a shows the distribution of waiting times between *consecutive* electrons tunneling out of the right island. Unlike the correlation functions, this is an *exclusive* probability density, since no tunneling events of the same type are allowed during the waiting time. Electron waiting times are typically hard to measure, since they require nearly perfect detectors, unlike the $g^{(2)}$-functions. The probability to observe a short waiting time is low, since only one electron at the time can tunnel out of the island. After having reached its maximum, the distribution falls off as it becomes exponentially unlikely to wait a long time for the island to be refilled and the next electron can tunnel out. These measurements agree well with the theoretical expectation

$$\mathcal{W}(\tau) = \frac{1}{\langle\tau\rangle u}\left(e^{-\gamma(1-u)\tau/2} - e^{-\gamma(1+u)\tau/2}\right), \quad (3)$$

which in addition to the inverse correlation time from Eq. (1) also contains the average waiting time between tunneling events, $\langle\tau\rangle = 1.2$ s, which enters through the parameter $u = \sqrt{1 - 4/(\gamma\langle\tau\rangle)}$. At short times, $\gamma\tau \ll 1$, the waiting-time distribution is linear in time, $\mathcal{W}(\tau) \simeq \gamma\tau/\langle\tau\rangle$, just as the $g^{(2)}$-function. By contrast, at longer times, where the $g^{(2)}$-function flattens out, it describes the small probabilities to observe long waiting times.

Turning to the cross-waiting-time distribution in Fig. 3b, we consider here the waiting time between an electron tunneling out of the left island followed by the tunneling of an electron out of

the right island. These experimental results are well-captured by a detailed theoretical analysis, which yields the expression

$$\mathcal{W}_x(\tau) = \frac{\eta_0}{2}\left[\sqrt{\frac{2}{\pi}}\frac{e^{-\frac{\tau^2}{2\sigma_D^2}}}{\sigma_D} + \mathcal{W}(\tau)\right] + (1 - \eta_0)\mathcal{W}_0(\tau), \quad (4)$$

where the first term in the brackets arises from the correlated two-electron processes, happening with the weight $\eta_0/2$, where $\eta_0 \simeq 0.36$, such as processes where an electron from each island tunnels into the superconductor to form a Cooper pair. The bracket also contains the distribution $\mathcal{W}(\tau)$ from Eq. (3), corresponding to the waiting time between a two-electron process and a subsequent transition on the right island. The last term in Eq. (4) originates from tunneling processes from the right island into the superconductor, which are weakly correlated with the left island and obey a Poissonian waiting-time distribution $\mathcal{W}_0(\tau) \propto e^{-\gamma(1-u)\tau/2}$ for long times. Importantly, the exponential dependence at long times allows us to characterize the storage time of the split electron pairs on the islands, i.e., the typical duration from ② to ③ in Fig. 2a. We note that the storage time ($\sim\gamma^{-1}$) is more than two orders of magnitude longer than the detection time of the splitting process ($\sim\sigma_D$).

## Discussion

We have observed the splitting of individual Cooper pairs in real time and thereby enabled the identification and storage of single split Cooper pairs, a procedure which is not possible with conventional current and noise measurements. As such, our work paves the way for a wide range of future experimental developments. Specifically, by embedding the single-electron detectors in a radio-frequency circuit, it should be possible to read out the charge states in a few microseconds[27,38], i.e., on a timescale that is much shorter than the spin lifetime and coherence time of semiconductor quantum dots, which typically exceed hundred microseconds[39,40]. In addition to the fast detection, the other key ingredient to probe the entanglement of the electrons is to detect the spin states of the split electrons. The readout methods of spin qubits are based on either the energy splitting of the spin states or spin-blockade, which would readily provide the required techniques[39,40]. Alternatively, ferromagnetic leads may be used to determine the spin state of the electrons[41,42]. Thus, by combining the fast charge detection with the spin readout, one may witness and certify the quantum entanglement of the split Cooper pairs[43,44]. Ultimately, several Cooper pair splitters may be combined to realize the first rudimentary quantum information algorithms using split Cooper pairs. Interestingly, our real-time

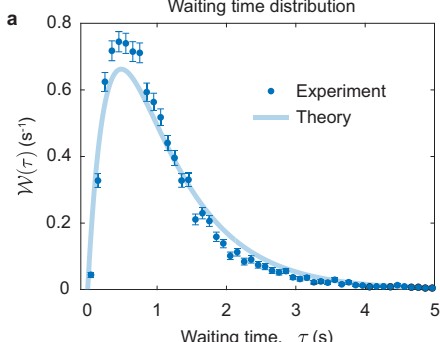

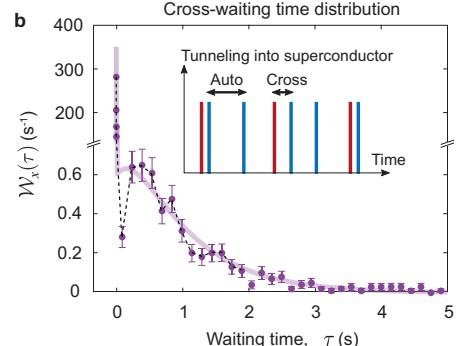

**Fig. 3 Waiting-time distributions $\mathcal{W}$. a** Waiting times $\tau$ between single-electron tunneling out of the right island. The theory curve given by Eq. (3) agrees well with the measurements. **b** Waiting times $\tau$ between single-electron tunneling out of the left island followed by single-electron tunneling out of the right island. Note that the peak at the short times is more than two orders of magnitude higher than the long time data. The experimental data is well-captured by Eq. (4). The dashed line serves as a guide to the eye. The inset illustrates the waiting times between tunneling events of the same or different types. The blue lines correspond to tunneling out of the right island, while the red ones correspond to the left island. The error bars mark $1\sigma$ confidence intervals.

detection scheme of crossed events would also enable us to discriminate different types of processes based on their statistics. For example, the instantaneous Cooper pair splitting events generate a sharp detection-limited peak as presented here, whereas the spin selective tunneling is discussed in refs. [41,42] would yield a tunneling time broadened peak that could be probed with the correlation function measurements. In conclusion, our work opens up avenues for experiments using entangled spins, and it thereby enables future quantum technologies based on entangled electrons in on-chip solid-state devices.

## Methods

**Device fabrication**. The device was fabricated on a silicon substrate with a 300-nm thermal silicon oxide layer on top. The fabrication started with a 2/30/2-nm-thick patterned Ti/Au/Ti layer (yellow in Fig. 1a) acting as a ground plane to filter out stray high-frequency noise. Then, a 10-nm-thick chromium strip (the green line running in 45° angles in the figure) was patterned and deposited to obtain a capacitive coupling between the islands and the detectors. Next, a 40-nm-thick AlOx layer was grown with atomic layer deposition to electrically insulate the ground plane and the coupler from the rest of the structures. Finally, electron beam lithography and shadow mask deposition were used to make the Cooper pair splitter and the charge detectors. In the last step, four metal layers were deposited at different angles with two oxidation steps in-between. First, the superconducting lead (in blue), 20-nm-thick Al, was formed. Next, the aluminum surface was oxidized to obtain tunnel barriers for the splitter. After that, the normal-state islands of 25-nm-thick copper (in orange) were deposited, completing the splitter structure. As a third layer, a second 30-nm-thick aluminum film was deposited followed by oxidation to produce the tunnel barriers for the detectors. The fabrication was then completed with the deposition of the 80-nm-thick copper leads of the detectors. The four-layer processing allows for independent tuning of the barrier transparencies for the splitter and the detectors. Fabricating the splitter first ensured that the aluminum reservoir is not directly connected to any of the extra non-operational shadow parts (seen e.g., in the insets showing the detectors), thus preventing any proximity effect. Also, the normal-state islands of the splitter are designed so that the second aluminum layer does not overlap with them until several micrometers away from the tunnel junctions. The detectors use the inverse proximity effect to suppress the superconductivity of the small aluminum patches connected directly to the first, 25-nm-thick Cu layer to obtain nearly normal metallic charge detectors.

The tunnel junctions in the Cooper pair splitter were made with a distance of $l = 100$ nm. This distance was chosen to be shorter than the coherence length $\xi = 200$ nm of the superconducting aluminum, hence allowing for a finite rate of Cooper pair splitting, which otherwise would be exponentially suppressed as $\exp(-l/\xi)$. By contrast, single-electron tunneling between the superconductor and the islands is suppressed as $\exp(-\Delta/k_B T)$, where $\Delta = 200$ μeV is the superconducting gap and $T = 50$ mK $\ll \Delta/k_B$ is the electronic temperature. Two-electron processes were thus the dominant charge transfer mechanism between the superconductor and the islands. The filtering with the ground plane was paramount for obtaining the suppression as stray radiation causes excess single-electron tunneling. We note that measurements with the superconducting electrode in the normal state were not performed, since the superconducting gap suppresses the sequential tunneling rates, which would otherwise exceed the detector bandwidth.

**Experiments**. All experimental results presented here were obtained at the base temperature of a dry dilution refrigerator with an electronic temperature of 50 mK. The detectors were biased with 200 μV, and the currents were measured with a digitizer after amplifying the signals with standard room temperature current preamplifiers with 1 kHz bandwidth. Voltages were applied to the gates to tune the charge detectors to a charge-sensitive operation point and to tune the populations of the normal-state islands so that the two lowest-lying states in Fig. 2a had equal probabilities. Time traces of the two detector currents were simultaneously recorded with a multichannel analog-to-digital converter at a sampling rate of 20 kHz, while adjusting the gate voltages in-between to compensate for slow drifts in the detector operation point or in the occupations of the two lowest charge states. The adjustment was done in a feedback loop by measuring a 60-s-long time trace and extracting population probabilities from it. The electronics and measurement configuration were made identical on both detector sides to minimize timing differences.

**Data analysis**. The 60-s-long time traces were digitally filtered through a low-pass filter with a cutoff frequency of 200 Hz which sets the detector rise times to about 4 ms. All the results presented in our paper were obtained by analyzing these filtered data. Time traces with the populations at the two lowest-lying charge states deviating considerably from each other or with the detector currents drifting so that it became difficult to identify charge states in the detector signal were excluded. The exact procedure and criteria are given in Supplementary Note 1. We then identified all tunneling events and the instances when they happened. These instances directly yield the experimental correlation functions in Fig. 1b and c and the waiting-time distributions in Fig. 3a and b. In addition to the instances, the theory curves require the detector broadening as an input parameter which was obtained from the current noise of the detectors and their slew rate.

**Theory**. The system is described by a rate equation, $\frac{d}{dt}||p(t)\rangle\rangle = \mathbf{L}||p(t)\rangle\rangle$, where $|p(t)\rangle\rangle$ is a column vector containing the probabilities of being in the different charge states of the islands, and the rate matrix $\mathbf{L}$ contains the tunneling rates. The off-diagonal elements of $\mathbf{L}$ are given by the transition rates between the different charge states, while the diagonal elements contain the total escape rate (with a minus sign) from each charge state. In addition, we introduce a jump operator, $\mathbf{J}_\alpha$, describing transitions of type $\alpha$. The $g^{(2)}$-function for processes of type $\alpha$ and $\beta$ can then be obtained as

$$g^{(2)}(\tau) = \frac{\langle\langle \mathbf{J}_\beta e^{\mathbf{L}\tau}\mathbf{J}_\alpha\rangle\rangle}{\langle\langle \mathbf{J}_\beta\rangle\rangle\langle\langle \mathbf{J}_\alpha\rangle\rangle}\theta(\tau) + \frac{\langle\langle \mathbf{J}_\alpha e^{-\mathbf{L}\tau}\mathbf{J}_\beta\rangle\rangle}{\langle\langle \mathbf{J}_\beta\rangle\rangle\langle\langle \mathbf{J}_\alpha\rangle\rangle}\theta(-\tau) + \frac{\langle\langle \mathbf{J}_{\alpha\beta}\rangle\rangle}{\langle\langle \mathbf{J}_\beta\rangle\rangle\langle\langle \mathbf{J}_\alpha\rangle\rangle}\delta(\tau),$$

(5)

where $\langle\langle \mathbf{A}\rangle\rangle \equiv \langle\langle \bar{0}|\mathbf{A}|p_s\rangle\rangle$ denotes the expectation value with respect to the steady-state fulfilling $\mathbf{L}|p_s\rangle\rangle = 0$, the vector representation of the trace operation is denoted as $\langle\langle \bar{0}|$, and $\theta(\tau)$ is the Heaviside step function. The $g^{(2)}$-function yields the normalized joint probability of detecting a transition of type $\alpha$ at some time and a transition of type $\beta$ at a time $\tau$ later. The last term in Eq. (5) accounts for the instantaneous correlations between electrons belonging to the same two-electron process, described by a jump operator $\mathbf{J}_{\alpha\beta}$. These correlations are convolved with a Gaussian distribution to take into account the timing jitter of the detectors. The $g^{(2)}$-function approaches unity on timescales over which the correlations vanish.

The waiting-time distribution can be obtained as

$$\mathcal{W}(\tau) = \frac{\langle\langle \mathbf{J}_\beta e^{(\mathbf{L}-\mathbf{J}_\beta)\tau}(\mathbf{J}_\alpha - \mathbf{J}_{\alpha\beta})\rangle\rangle}{\langle\langle \mathbf{J}_\alpha\rangle\rangle} + \eta_0\delta(\tau),$$

(6)

with $\eta_0 \equiv \frac{\langle\langle \mathbf{J}_{\alpha\beta}\rangle\rangle}{\langle\langle \mathbf{J}_\alpha\rangle\rangle}$, which, just as for the $g^{(2)}$-function, includes the instantaneous correlations of two-electron processes. The waiting-time distribution yields the probability density to observe a waiting time $\tau$ from a transition of type $\alpha$ has occurred until the first subsequent transition of type $\beta$ takes place. Similar to the $g^{(2)}$ function, we use convolution with a Gaussian distribution to describe the timing jitter of the detectors. As a probability distribution, the waiting-time distribution is normalized such that $\int_0^\infty d\tau \mathcal{W}(\tau) = 1$.

## Data availability
The data that support the findings of this study are available from the corresponding authors upon reasonable request.

## Code availability
The code to analyze the experimental data and to compute the theoretical results is available from the corresponding authors upon reasonable request.

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

## Acknowledgements

We thank J.P. Pekola and P. Samuelsson for fruitful discussions. This work was supported financially by the QuantERA project 2D hybrid materials as a platform for topological quantum computing, Swedish National Science Foundation, NanoLund, and the Academy of Finland (project numbers 308515 and 331737). F.B. acknowledges support from the European Union's Horizon 2020 research and innovation program under the Marie Skłodowska-Curie grant agreement number 892956. F.B., E.T.M., and C.F. acknowledge the support by the Academy of Finland through the Finnish Centre of Excellence in Quantum Technology (project numbers 312057 and 312299). We acknowledge the provision of facilities by Aalto University at OtaNano—Micronova Nanofabrication Centre.

## Author contributions

The experiment was carried out by A.R. and V.F.M. The devices were fabricated by E.T.M. The theory was developed by F.B. and C.F. All authors contributed to the discussion and analysis of the results and the writing of the manuscript.

## Funding

## Competing interests

The authors declare no competing interests.
