## [Peer Review File · Nature Communications]

REVIEWER COMMENTS

Reviewer #1 (Remarks to the Author):

The central finding of this paper, as encapsulated in Fig. 2 of the main text, if it does indeed show what it claims to show, is certainly worthy of publication in Nature Communications. However, there are many questions about the geometry of the device and the execution of the experiment, and it is not clear what the authors hope to prove by their analysis, which in my opinion serve only to obfuscate the results. At the least, the paper requires a complete rewrite.

Let me first focus on the experimental aspects of the paper. First, a general opinion: since this is primarily an experimental paper, key details of the experiment should be given in the main text, and not buried somewhere in the supplementary materials. For example, simple details of the electronic setup are not given: Is some part of the actual sample grounded? What is the temperature? (there is mention in the Methods that the electronic temperature is 40 mK...is this for all measurements reported in the paper, and how is this determination made?). Also, in this experiment, to borrow a phrase, timing is everything, but key timing details of how the data are taken are given only in the supplementary or not given at all. The statements are sometimes contradictory, and cast doubt on the analysis. I explain in more detail about these issues below.

The experimental setup appears at first sight to be straightforward. From my reading of the paper, the device consists of a superconductor connected to two normal metal islands through tunnel junctions, but otherwise isolated. There is only one mention in passing in the supplementary that the superconductor is grounded—this needs to be stated more prominently. The charge on each normal island is detected by capacitively coupling each normal island to a SET. In the figure, the normal islands are colored orange, but the device geometry that capacitively couples these normal islands to each SET is not clear.

In the supplementary, it is mentioned that each normal island is maintained at the charge neutrality point. I could not find anywhere in the main paper or supplementary how this was achieved. After many re-reads, I finally figured out that the two electrodes shown at the top right and left of Fig. 2a are gates used to control the charge state of the normal islands. This is not mentioned anywhere in the main text or the supplementary. All these key details need to be in the main text of the paper.

But this raises an important question that the authors need to address. Changing the gate voltage changes the potential of each island with respect to the superconductor, and with respect to each other. Based on some earlier theoretical predictions, this should change the probability of crossed andreev reflection/elastic co-tunneling (see for example, G. Falci et al 2001 EPL 54 255). Have the authors considered this?

Now regarding the time dependent data. I was hard put to find all of the relevant experimental timing information, none of which was in the main text, where it should be as this paper is all about timing, but scattered in various places in the supplementary. This is what I gathered about the timing measurements. The data are taken at a sampling rate of 20 kHz, which corresponds to a point every 50 usec. I presume that the data points on both channels are taken simultaneously using something like simultaneous sampling ADCs, otherwise the whole premise of the paper is moot. The specifications of this instrumentation needs to be given in the main text. There is some mention of "jitter" in the supplementary, but no numbers are given and I don't know what this means. Next, the authors state that the data are digitally filtered with a low-pass filter with a cutoff frequency of 200 Hz. (The authors mention that this restricts the detector rise times to 4 msec, which seems shorter than it should be, but about the right range.) This means that any correlations below a time scale of 5 ms are

meaningless. Thus, I immediately do not understand Fig. 1c, which appears to show correlations dying off in fractions of a msec. Before starting any analysis, the authors need to clarify the data on which they are performing the analysis. Is it the filtered data? (it does not seem to be, at least for the cross correlation) Is it the raw data? (if so, how is the noise taken into account?).

Now to the analysis. The authors use two measures in their analysis, auto-correlation and cross-correlation, but the convoluted way they do the analysis left me completely confused and with no confidence in their results. First, calculating an autocorrelation of any data set numerically is a fairly straightforward matter. For data with noise, one usually sees a sharp peak near zero which corresponds to the autocorrelation of the noise, which can be taken care of most simply by filtering this high frequency noise in the frequency domain, and then doing an inverse fourier transform. Now the time series data shown in Fig. 2a looks very much like a two-level fluctuator, no doubt because the gates have been tuned to the charge neutrality point. We know for many years now from the Dutta-Horn analysis that the Fourier time of the autocorrelation is a Lorentzian that gives the characteristic fluctuation and the autocorrelation should look like $1 - g_2(\tau)$ shown in Fig. 1b, presumably because of the way authors define $g_2(\tau)$. The characteristic time of the system may be related to the tunneling rates, but it is not clear to me what the complicated analysis based on histograms etc detailed in the supplementary provides that is not given by the simple standard analysis described above.

Analysis of the cross-correlation between the two detectors to pull out the crossed Andreev and elastic co-tunneling events is obviously more difficult, but the analysis in the supplementary again does not inspire confidence. I have already mentioned the issue with the filter time constant above. In spite of this, the authors mention considering correlations after a time scale of 170 usec (again, no explanation or justification given for this time scale). There is a whole discussion of noise and slew rates, all of which could be taken care of simply considering the FT of the raw cross-correlation data as discussed above for the autocorrelation. As to pulling out the CAR and EC events: I would anticipate that an actual measure would involve the cross-correlation as well as the individual autocorrelations..an analysis like this would be of real benefit to the community.

Finally, in the long theoretical analysis in the supplementary, I found no connection to the physics of cross Andreev reflection. It just seems a complicated way to fit the data.

To summarize, while I think the basic result of this work is worth publishing, the entire manuscript needs to be reworked before it is ready for publication. In my opinion, a far more powerful statement can be made by simply showing a series of simultaneous time traces as shown in Fig. 2, with simple statistics on how often various events occur. Even more powerful would be to show that crossed Andreev reflection and elastic cotunneling disappear when the superconductor is no longer superconducting, which can easily be done in the same device by applying a large enough magnetic field.

Reviewer #2 (Remarks to the Author):

The paper "Real-time observation of Cooper pair splitting showing strong non-local correlations" by A. Ranni et al. reports on charge correlation measurements in a Y-junction device made with a central superconducting electrode and two outer metallic contacts. Such devices have attracted a lot of attention in the last years due to their potential for the use of the spin-entangled singlet state as a resource for quantum information. Transport measurements as well as microwave measurements have been carried out in nanowire, nanotube or graphene based systems alike. None of the

measurements could explore the real-time correlations. In the spirit of quantum optics experiments, it is crucial to perform time domain correlations. This is what is achieved in the present work, which makes this paper very interesting. I am ready to recommend publication in Nature Communications, but I would like the authors to consider the following comments :

1. The effect of Coulomb interactions is important for the Cooper pair splitting process. However, it has been shown some time ago that interactions can generate positive cross correlations in similar setups but without the superconductor (see A. Cottet et al. PRL 92, 206801 (2004) and PRB 70 115315 (2004)). Although I appreciate that this was mainly for the zero frequency shot noise that this point was made, I would like the authors to discuss how their real time detection scheme enables discrimination from these processes.

2. Have the authors tried to apply a magnetic field to study the normal state ? It was not fully clear to me.

3. In the present work, the timescales probed by the experiment are very slow, so the coherence of Cooper pair splitting cannot be assessed. I think it would be nice if the authors could put a bit more perspective on the technical evolution which should be done in order to probe the system with the relevant timescales for sensing entanglement and to identify the spin states for the correlated processes.

4. I find that the description of the setup is a bit too sketchy. I think it is important to describe in more details in the main text the setup, both the layout and the electronics.

Response to Reviewers:

We thank both Reviewers for the positive feedback on our manuscript. Reviewer 1 finds that our central findings are “certainly worthy of publication in Nature Communications”, and Reviewer 2 writes that the real-time detection of Cooper pair splitting “makes this paper very interesting. I am ready to recommend publication in Nature Communications.” In addition, the Reviewers are raising the following issues: Reviewer #1 is concerned about our measurements of cross-correlations on a timescale which is shorter than the detector rise time. However, as we explain below, it is indeed meaningful and possible to perform such measurements. Reviewer #2 is in favor of publishing our manuscript, once we have addressed a couple of issues.

Below, we respond to all concerns, and we attach the revised manuscript.

Response to Reviewer #1:

Reviewer #1: The central finding of this paper, as encapsulated in Fig. 2 of the main text, if it does indeed show what it claims to show, is certainly worthy of publication in Nature Communications. However, there are many questions about the geometry of the device and the execution of the experiment, and it is not clear what the authors hope to prove by their analysis, which in my opinion serve only to obfuscate the results. At the least, the paper requires a complete rewrite.

Our response: We thank the Reviewer for the careful reading of our manuscript and for providing us with several useful comments and suggestions that we respond to below. Let us first reassure the Reviewer that we indeed observe what we claim, namely the splitting of individual Cooper pairs. However, to substantiate this claim, a careful statistical analysis is required. To this end, we show in Figs. 1b and 1c results for the auto and cross-correlation functions, which are similar to what is often measured for photons in quantum optics. This statistical analysis allows us to conclude that two nearly simultaneous tunneling events, as the ones in Fig. 2a, with 99% fidelity correspond to the splitting of a Cooper pair rather than just being two uncorrelated single-electron events. Thus, the statistical analysis in Fig. 1 is an important element in support of our claim. Here, let us also comment on an issue that is raised by the Referee below, namely, if it is possible to measure cross-correlations on a timescale, which is shorter than the rise time of the detectors. This is a critical issue for the results in Fig. 1c, and as we clarify below and in the updated manuscript material, it is indeed possible to measure cross-correlations on such short timescales.

We would also like to point out that the correlation measurements, in particular Fig. 1c, are noteworthy results by themselves, and such measurements are important tools for statistical characterizations. In the following responses, we clarify the concerns about the cross-correlation measurements, and hence we do not see a need to completely rewrite the manuscript. Instead, we have added further details of our analysis, which indeed is sound and valid.

Corresponding changes: We have revised our manuscript so that the electrical circuitry is indicated in Fig. 1a and described in the main text. We have also added a new paragraph and a figure in the Supplemental information (see Fig. 3b) to address the fast time resolution of the cross-correlation function below the detector rise time.

Reviewer #1: Let me first focus on the experimental aspects of the paper. First, a general opinion: since this is primarily an experimental paper, key details of the experiment should be given in the main text, and not buried somewhere in the supplementary materials. For example, simple details of the electronic setup are not given: Is some part of the actual sample grounded? What is the temperature? (there is mention in the Methods that the electronic temperature is 40 mK...is this for all measurements reported in the paper, and how is this determination made?). Also, in this experiment, to borrow a phrase, timing is everything, but key timing details of how the data are taken are given only in the supplementary or not given at all. The statements are sometimes contradictory, and cast doubt on the analysis. I explain in more detail about these issues below.

Our response: We thank the Reviewer for pointing out these issues, and we now provide the additional details about the measurements and the experimental setup.

Corresponding change: In the revised main text, we now state the conditions under which the experiment is performed, and we provide further details of the measurements.

Reviewer #1: The experimental setup appears at first sight to be straightforward. From my reading of the paper, the device consists of a superconductor connected to two normal metal islands through tunnel junctions, but otherwise isolated. There is only one mention in passing in the supplementary that the superconductor is grounded—this needs to be stated more prominently. The charge on each normal island is detected by capacitively coupling each normal island to a SET. In the figure, the normal islands are colored orange, but the device geometry that capacitively couples these normal islands to each SET is not clear.

Our response: Indeed, as the Reviewer is pointing out, the essential idea behind the experiment is rather simple and straightforward, and it is accurately described by the Reviewer above. Having said so, there are also many additional details and underlying technicalities, and we have tried to strike a proper balance between technical details for the expert reader as well as simple and clear messages for the general readership of the journal. However, we agree that some information might belong better in the main text than in the supplemental material.

Corresponding change: As mentioned above, we have updated Fig. 1, so that the electrical circuitry of our experimental setup is clarified, including the grounding of the superconductor. The coupling strip is now highlighted with green in the micrograph picture, and it is described in the caption.

Reviewer #1: In the supplementary, it is mentioned that each normal island is maintained at the charge neutrality point. I could not find anywhere in the main paper or supplementary how this was achieved. After many re-reads, I finally figured out that the two electrodes shown at the top right and left of Fig. 2a are gates used to control the charge state of the normal islands. This is not mentioned anywhere in the main text or the supplementary. All these key details need to be in the main text of the paper.

Our response: Yes, the two electrodes in the upper left and right corners of Fig. 1a are the gate electrodes that we use to control the charge state of the two islands. We now clarify that in the revised manuscript.

Corresponding change: We have added the gate voltages in Fig. 1a and included a sentence in the main text to describe that we adjust the gate voltages to maintain the two lowest charge states at equal energies alongside with two additional references, where the feedback-loop is applied and described (see Refs. 35 and 36). The feedback loop, which adjusts the gate voltages to compensate for drifts, is now briefly described in the Methods sections.

Reviewer #1: But this raises an important question that the authors need to address. Changing the gate voltage changes the potential of each island with respect to the superconductor, and with respect to each other. Based on some earlier theoretical predictions, this should change the probability of crossed andreev reflection/elastic cotunneling (see for example, G. Falci et al 2001 EPL 54 255). Have the authors considered this?

Our response: Yes, these considerations are indeed important, and they formed the basis for the design of our device and the experiment. Using the gate voltages, we can control the potential of the islands with respect to the superconductor and thereby change the tunneling rates. By having the lowest charge states degenerate, there is no energy cost involved in crossed Andreev reflections and for elastic cotunneling processes between the islands. However, for the detection of the tunneling processes and the measurements of the correlation functions, the energy dependence of the tunneling rates is not essential: The detection of the individual tunneling processes can be done as long as the tunneling rates are not too large, and our measurements directly show that this is the case.

Corresponding change: We now briefly mention in the main text that maintaining the two lowest charge states at degeneracy ensure that there is no energy cost associated with processes between them.

Reviewer #1: Now regarding the time dependent data. I was hard put to find all of the relevant experimental timing information, none of which was in the main text, where it should be as this paper is all about timing, but scattered in various places in the supplementary. This is what I gathered about the timing measurements. The data are taken at a

sampling rate of 20 kHz, which corresponds to a point every 50 usec. I presume that the data points on both channels are taken simultaneously using something like simultaneous sampling ADCs, otherwise the whole premise of the paper is moot. The specifications of this instrumentation needs to be given in the main text. There is some mention of “jitter” in the supplementary, but no numbers are given and I don’t know what this means. Next, the authors state that the data are digitally filtered with a low-pass filter with a cutoff frequency of 200 Hz. (The authors mention that this restricts the detector rise times to 4 msec, which seems shorter than it should be, but about the right range.) This means that any correlations below a time scale of 5 ms are meaningless. Thus, I immediately do not understand Fig. 1c, which appears to show correlations dying off in fractions of a msec. Before starting any analysis, the authors need to clarify the data on which they are performing the analysis. Is it the filtered data? (it does not seem to be, at least for the cross correlation) Is it the raw data? (if so, how is the noise taken into account?)

Our response: In the auto-correlation measurements, the detector rise-time indeed imposes a limitation, since the detector needs to “recover” before the next event can be observed. We refer to this time scale as the dead time. As shown in the Supplementary information, we see this limitation in Fig. 4, where a dip with the width of the detector rise time appears at short times. However, the cross-correlation measurements do not have this limitation. This is because we use two detectors, one for each event and hence the dead time of the detectors is not a limiting factor. Instead, it is rather the noise in the timing of the two detectors that matters. These fluctuations are known as jitter, and we thank the Referee for pointing out that these features are not obvious. We have thus extended the description of the correlation function measurements extensively in to the supplemental material with additional experimental data and a description of how the two detection methods differ. We show now directly based on experimental data how the cross-correlations can be measured on a time scale, which is shorter than the detector rise time. We hope that these additions clarify these issues and makes it clear that our results indeed are sound and correct. For all of our analysis, we have used data, for which high-frequency fluctuations have been filtered out.

Corresponding change: The timing details including the specifications of the instrumentation are now all provided in the Methods section of the manuscript, including details of the sampling and the sampling rate. We also mention that the time traces are measured simultaneously and that we use the filtered data for all of the analysis. We have also added content to the supplementary material to demonstrate the sub-risetime detection resolution of the cross-correlations including a new figure with experimental data (see Fig. 3b in the supplementary information) that directly shows how this works. We also briefly describe the fast cross-correlation measurements in the main text.

Reviewer #1: Now to the analysis. The authors use two measures in their analysis, auto-correlation and cross-correlation, but the convoluted way they do the analysis left me completely confused and with no confidence in their results. First, calculating an autocorrelation of any data set numerically is a fairly straightforward matter. For data with noise, one usually sees a sharp peak near zero which corresponds to the autocorrelation of the noise, which can be taken care of most simply by filtering this high frequency noise in the frequency domain, and then doing an inverse fourier transform. Now the time series data shown in Fig. 2a looks very much like a two-level fluctuator, no doubt because the gates have been tuned to the charge neutrality point. We know for many years now from the Dutta-Horn analysis that the Fourier time of the autocorrelation is a Lorentzian that gives the characteristic fluctuation and the autocorrelation should look like $1 - g_2(\tau)$ shown in Fig. 1b, presumably because of the way authors define $g_2(\tau)$. The characteristic time of the system may be related to the tunneling rates, but it is not clear to me what the complicated analysis based on histograms etc detailed in the supplementary provides that is not given by the simple standard analysis described above.

Our response: We believe that the questions about the cross-correlation measurements above may have caused some concerns about our experimental results. However, let us start by stressing that our analysis of the experimental data in fact is very simple, *both for auto-correlation and cross-correlations*: Once we detect a tunneling event using one of the detectors, we simply determine, if another tunneling event happens in a short time window around a later time τ . The auto-correlations are obtained by considering tunneling events in the same island, while the cross-correlations consider tunneling events in different islands. The island at which the second event is considered is the only difference between the two measurements. Otherwise, the analysis is the same in both cases.

Regarding the characteristic time scale of the g_2 -function in Fig. 1b, the Reviewer is correct that it can be related to the tunneling rates, and that is in fact how we determine the value of $1/4.5$ s that enters Eq. (1) of the main text. However, as shown by our theoretical analysis in the supplemental material, the expression for this characteristic time scale is a complicated function of the tunneling rates, see Equations (6,7) of the Supplementary information. The reason for this is that the both islands have several accessible charge states, implying that the system is not just a two-level fluctuator. Still, as our theoretical analysis shows, the resulting g_2 -function takes the same form as for a two-level fluctuator, albeit with a characteristic time scale that is a complicated function of all tunneling rates.

We acknowledge that one may analyze the data from many different angles, including the approach suggested by the Reviewer. However, for our purposes, we are convinced that the standard approach to measurements of g_2 -functions in quantum optics is the appropriate way to analyze our experiment. Also, as we discuss below, our experimental results are supported by a theoretical model with no free parameters, and the good agreement between theory and experiment makes us further confident in our analysis of the experimental results.

Corresponding change: We have clarified in the supplemental material the key idea how we determine the correlation functions and we stress the similarity between the auto- and cross-correlation measurements as well as explicitly state the only difference between the two.

Analysis of the cross-correlation between the two detectors to pull out the crossed Andreev and elastic co-tunneling events is obviously more difficult, but the analysis in the supplementary again does not inspire confidence. I have already mentioned the issue with the filter time constant above. In spite of this, the authors mention considering correlations after a time scale of 170 usec (again, no explanation or justification given for this time scale). There is a whole discussion of noise and slew rates, all of which could be taken care of simply considering the FT of the raw cross-correlation data as discussed above for the autocorrelation. As to pulling out the CAR and EC events: I would anticipate that an actual measure would involve the cross-correlation as well as the individual autocorrelations..an analysis like this would be of real benefit to the community.

Our response: This comment concerns three different points that we now address in turn:

(1) To begin with, we stress that it is not more difficult to determine the cross-correlations than the autocorrelations, and they are determined in a similar way as discussed in the response above.

(2) The discussion about the slew rates and noise in the detector is made in the supplemental material to explain the width of the peak in Fig. 1c. The width is given directly by the timing accuracy arising from the detector jitters.

(3) The 170 usec is the bin size that we use for determining the correlation functions. It sets the limit for the time resolution of the distributions and should thus be kept as short as possible. At the same time, the bin size needs to be large enough to collect a reasonable number of counts for each bin, and we have chosen a bin size that balances between these two requirements: It is small enough to resolve the peak structure in Fig. 1c of the main text and large enough for us to collect sufficient statistics with small error bars as illustrated in the figure. Choosing a bin size in this manner is a standard procedure.

However, we thank the Reviewer for bringing up the 170 usec timescale, which led us to think about the choice of bin size more carefully. The sampling rate of 20 kHz gives a time discretization of 50 usec in the detector signals as the Reviewer mentioned earlier. Thus, when looking at time differences between events, the possible results are integer multiples of 50 usec. The ideal way to choose the time window is to make it commensurate with the time separation of the time points. This ensures that each bin has equal number of time points in it and removes a scatter that may arise from having some bins that have one time point more or less than the other bins depending on how the time points fall into the window or out of the time window. We thus changed the time window (i.e. bin size) to 150 usec so that each bin has exactly three time points in it. This improvement led to a correction of the data point at $\tau = 170$ usec in Fig. 1b that was previously too high. This bin had indeed four time points instead of the three that the neighboring data points had. We therefore changed to the commensurate bin size. Otherwise, this improvement made no changes to the resulting figure apart from minor changes one data point.

Corresponding changes: We updated the experimental data of Fig. 1c so that the bin size is now 150 usec. The theoretical results are unaltered. The choice of bin size is now explained in the supplementary information in the section “Measurements of the correlation functions”. In the same section, we also explain that the jitter causes the broadening of the peak, and we have extended the supplementary information to introduce the time jitter and related concepts with more detail and in a more pedagogical way. The broadening of the peak is given by Equation (9) in the supplementary information.

Reviewer #1: Finally, in the long theoretical analysis in the supplementary, I found no connection to the physics of cross Andreev reflection. It just seems a complicated way to fit the data.

Our response: Let us start by stressing that the theoretical analysis is not simply a fit-to-the-data. On the contrary, we use a rate-equation model that includes all accessible states in the system and all transition rates between them are directly obtained from the measured time traces. As such, our starting point is a theoretical model with *no* free parameters. Based on our theoretical model, we then evaluate the auto and cross correlation functions and the distributions of waiting times, and we find excellent agreement between theory and experiment. This agreement is an important indicator of the robustness and accuracy of our measurements, and the theoretical model is also useful for the analysis of the experiment. Here, the crossed Andreev reflections play an important role as one of the possible transfer processes in the system. In fact, these processes, together with the timing jitter of the detectors, are crucial for the features of the cross-correlation function, which is one of our main results.

Corresponding changes: In the revised manuscript, we have further clarified the connection between the theory and the experimental data.

Reviewer #1: To summarize, while I think the basic result of this work is worth publishing, the entire manuscript needs to be reworked before it is ready for publication. In my opinion, a far more powerful statement can be made by simply showing a series of simultaneous time traces as shown in Fig. 2, with simple statistics on how often various events occur.

Our response: We thank the Reviewer for the useful comments above, and it should now be clear that it indeed is meaningful to consider cross-correlation measurements on time-scales that are shorter than the detector rise time. It seems that this was a critical issue, which raised some concerns about the results in Fig. 1. With the above clarifications, the issues about the validity of these measurements have been resolved. Indeed, the results are accurate and remain the same as in the initial submission. In response to the comments by the Reviewer, we have clarified the concerns about the timing and the accuracy of our measurements. In the revised manuscript, we now describe how the cross-correlations measurements can be made with higher time resolution than the auto-correlations, and that the detector rise time does not affect the cross-correlation measurements.

We are happy to learn that the Reviewer finds the data of Fig. 2 worth publishing in Nature Communications. Our results for the correlation functions in Fig. 1 contributes to the impact of Fig. 2. The fact that the two detectors combined provide a time resolution, which is higher than the rise time of the individual detectors, was not discussed in the initial version of our manuscript, but we now have revised our manuscript accordingly.

Corresponding change: We have updated the manuscript to address all the points requested by the reviewer as discussed above. We additionally now point out in the supplemental material that the correlation function measurements are in fact a method based on analyzing how often various events occur. This is achieved by describing in depth in the updated supplemental material that the correlation function measurements count how often each event type occurs. The fact that the correlation measurements consider the number of events that occur was mentioned already briefly in the original main text with the phrase “*We now consider the conditional probability of observing a tunneling event between the superconductor and the right island at the time delay τ after a tunneling event between the super-conductor and the left island has occurred.*” We believe that this a sufficiently detailed description in the main manuscript, which explains the key idea of our cross-correlation measurements.

Reviewer #1: Even more powerful would be to show that crossed Andreev reflection and elastic cotunneling disappear when the superconductor is no longer superconducting, which can easily be done in the same device by applying a large enough magnetic field.

Our response: We agree that it would be compelling to repeat the experiment under conditions, where the aluminum part is no longer superconducting, for example, by applying a large magnetic field. However, in the normal state, the single-electron tunneling rates would dramatically increase, and they would become much larger than the detector bandwidth and hence prevent us from detecting the tunneling events in real time. Specifically, in the superconducting state, the single-electron processes are suppressed as $\exp(-\Delta/kT)$, where Δ (≈ 200 μeV) is the superconductor gap for aluminum. Thus, without this large gap, we could not perform the experiment.

Corresponding change: We added a sentence to the methods section to comment on this.

Response to Referee #2:

Reviewer #2: The paper "Real-time observation of Cooper pair splitting showing strong non-local correlations" by A. Ranni et al. reports on charge correlation measurements in a Y-junction device made with a central superconducting electrode and two outer metallic contacts. Such devices have attracted a lot of attention in the last years due to their potential for the use of the spin-entangled singlet state as a resource for quantum information. Transport measurements as well as microwave measurements have been carried out in nanowire, nanotube or graphene based systems alike. None of the measurements could explore the real-time correlations. In the spirit of quantum optics experiments, it is crucial to perform time domain correlations. This is what is achieved in the present work, which makes this paper very interesting. I am ready to recommend publication in Nature Communications, but I would like the authors to consider the following comments:

Our response: We thank the Referee for the careful reading of our manuscript and for providing us with the useful comments below. We are happy to learn that the Referee finds that our paper is highly interesting.

Reviewer #2: 1. The effect of Coulomb interactions is important for the Cooper pair splitting process. However, it has been shown some time ago that interactions can generate positive cross correlations in similar setups but without the superconductor (see A. Cottet et al. PRL 92, 206801 (2004) and PRB 70 115315 (2004)). Although I appreciate that this was mainly for the zero frequency shot noise that this point was made, I would like the authors to discuss how their real time detection scheme enables discrimination from these processes.

Our response: This is an interesting comment, and we expect that our detection scheme would indeed make it possible to distinguish Cooper pair splitting from the bunching effect discussed in the papers above due to their different statistical properties: While Cooper pair splitting leads to nearly simultaneous tunneling events into the two islands, the bunching effect mentioned above has a characteristic time difference between the events arising from the tunneling rates. For Cooper pair splitting, we observe a sharp peak at short times in the g_2 -function, limited by the measurement noise, while the processes described in the above works would also result in a peak in the cross-correlation function but with the width set by a characteristic tunneling timescale. If the tunneling time broadening is larger than the timing jitter, our detection approach would allow us to distinguish these processes, even without studying the other physical properties such as the gate dependence or magnetic field dependence of the rates.

Corresponding change: We now include the references pointed out by the Referee towards the end of the manuscript, when we discuss the possibilities of measuring the spin states using ferromagnetic electrodes.

Reviewer #2: 2. Have the authors tried to apply a magnetic field to study the normal state? It was not fully clear to me.

Our response: The same question was raised by Reviewer #1, but unfortunately it is not possible to carry out the experiment at high magnetic fields, since the rate of single-electron tunneling would become much larger than the bandwidth of our detectors.

Corresponding change: We have added a sentence about this limitation in the Methods section.

Reviewer #2: 3. In the present work, the timescales probed by the experiment are very slow, so the coherence of Cooper pair splitting cannot be assessed. I think it would be nice if the authors could put a bit more perspective on the technical evolution which should be done in order to probe the system with the relevant timescales for sensing entanglement and to identify the spin states for the correlated processes.

Our response: Indeed, we did not aim for observing the coherence of the split Cooper pairs, and this is certainly an interesting direction for future work. Towards the end of the manuscript, we gave an outlook on some of the requirements for detecting the coherence of the split Cooper pairs and potentially even the non-local entanglement of the spins. We believe that such experiments are within reach based on state-of-the-art semiconductor spin qubits that maintain their coherence on us-timescales combined with fast radiofrequency charge readout on the same timescales. Based on this comment by the Reviewer, we have somewhat extended the discussion of these perspectives. At the same time, we are hesitant to speculate too much and are thus keeping the discussion short.

Corresponding change: We have extended the perspectives section to describe the requirements for two key elements: long enough coherence times and fast enough detection.

Reviewer #2: 4. I find that the description of the setup is a bit too sketchy. I think it is important to describe in more details in the main text the setup, both the layout and the electronics.

Our response: We thank the Reviewer for this comment and note that the same issue was pointed out by Reviewer #1. We have thus revised our manuscript and Fig. 1 accordingly.

Corresponding change: We have added further details about the setup and the electronics, in particular in Fig. 1.

List of changes: We are attaching the revised manuscript with changes indicated in green.

REVIEWER COMMENTS

Reviewer #1 (Remarks to the Author):

The authors have made substantial elaborations, particularly in the description of the experimental setup that significantly improve the manuscript. I still believe that the theoretical description and analysis complicates the relatively straightforward result of the experiment, but I am willing to acknowledge that looking at the results from the quantum optics perspective might be valuable.

However, there is still a critical part of the analysis that is faulty, and this is with regard to the minimum time for measuring the cross-correlation functions. As the authors clarify in their response and the updated manuscript, all the analyses are performed on data filtered to give a rise time of 4 ms, including digitization of the signal by a simultaneously sampled data acquisition card. First, I do not understand why the noise of the signal should affect the jitter. The jitter should be determined by the specs of the card, perhaps by the jitter in the clock running the ADCs. It seems the authors are confusing uncertainty in the measurement of the amplitude of the signal with uncertainty in measuring the time, which does not make sense, at least to me.

More important, I do not see how one can get correlation data on time scales less than the filter time of the data. Assume, for simplicity, that a brick wall low pass filter is employed (although I assume that the authors use something more sophisticated). Then the Fourier transform of the cross-correlation is zero beyond a frequency of 200 Hz, which means that the frequency range over which the cross-correlation is calculated is also terminated at 200 Hz, so that the minimum time over which the discrete real time cross-correlation can be calculated is 4 ms or so. Anything less is not meaningful. I may be wrong, but I would like the authors to show mathematically why this is incorrect.

In the same vein, the authors now also give new information that the data are acquired using preamplifiers with a bandpass of 1 kHz before being digitized. I assume these are low pass filtered: again, I am not sure why the data are digitized at 20 kHz when the data being pre-filtered at 1 kHz?

I am also a bit confused by the discussion of the feedback loops used to keep the islands at the charge neutrality point. It seems that these are only used in between measurements to correct for drifts. Are they in operation during the measurements as well? If so, the frequency response of the feedback loops will also come into play in the analysis, unless the time scales are vastly different from the time scales of interest.

Since this is a central point of the paper, I believe it needs to be resolved before the paper is suitable for publication.

Reviewer #2 (Remarks to the Author):

The authors have revised their manuscript along the suggested lines from both the reviewers. They have in particular correctly addressed the sharp questions of reviewer #1. I recommend publication in Nature Communications.

We are happy to learn that Reviewer #2 now clearly recommends publication of our manuscript, and the Reviewer also confirms that we have correctly addressed all concerns raised by both Reviewers. Reviewer #1 acknowledges that we have made substantial elaborations and significantly improved the manuscript. Still, despite our previous response and the corresponding revisions of the supplement material, the Reviewer is still concerned about the minimum time for measuring the cross-correlations, which should be 4 ms or more according to the Reviewer. In this regard, we would like to draw attention to the inset of Fig. 3b of the supplement, where it can clearly be seen that the difference between the time, when the full blue curve crosses the dashed blue line, and the time, when the red full curve crosses the red dashed line, is about 0.1 ms, which is much smaller than 4 ms. Thus, our claim is fully supported by our experimental observations.

In the response below, we reiterate this explanation. However, we see no need for further changes to our manuscript except an additional reference in the supplemental material explaining how amplitude fluctuations in the signal induce timing jitter. (We speculate that our quantum optics approach to the data analysis may be surprising to those who are more familiar with conventional solid-state approaches based on low-frequency noise correlators. However, that does not imply that anything should be wrong with our analysis.)

Thus, we believe that the current version of our manuscript should be ready for publication in Nature Communications.

We would like to thank you and the Reviewers for your efforts in reviewing our manuscript.

Yours sincerely,

Antti Ranni, Fredrik Brange, Elsa T. Mannila, Christian Flindt, and Ville F. Maisi

Response to Second Report of Reviewer #1:

Reviewer #1: The authors have made substantial elaborations, particularly in the description of the experimental setup that significantly improve the manuscript. I still believe that the theoretical description and analysis complicates the relatively straightforward result of the experiment, but I am willing to acknowledge that looking at the results from the quantum optics perspective might be valuable.

Our response: We thank the Reviewer for the careful reading of our response and the revised manuscript. Indeed, we are analyzing our experiment from a quantum optics perspective, which might be somewhat different from the typical solid-state approach of looking at low-frequency noise correlators. We speculate that this difference may have caused some confusion. Regarding our theoretical analysis, it leads to the simple and compact expressions in Eqs. (1-4), which to a very high degree capture our experimental results. As such, we believe that they are important, as they make us confident in our analysis of the experiment, and they also give insights into how the system parameters control the measurements.

Reviewer #1: However, there is still a critical part of the analysis that is faulty, and this is with regard to the minimum time for measuring the cross-correlation functions. As the authors clarify in their response and the updated manuscript, all the analyses are performed on data filtered to give a rise time of 4 ms, including digitization of the signal by a simultaneously sampled data acquisition card. First, I do not understand why the noise of the signal should affect the jitter. The jitter should be determined by the specs of the card, perhaps by the jitter in the clock running the ADCs. It seems the authors are confusing uncertainty in the measurement of the amplitude of the signal with uncertainty in measuring the time, which does not make sense, at least to me.

Our response: Timing jitter describes the fluctuations of the temporal position of pulses, in our case, the tunneling events. It is true that the timing of the electronics is one source for this as the Reviewer is pointing out. However, in our case, the detector current noise (*i.e.* the fluctuations in the signal) causes fluctuations in the time at which the event is detected.

The current noise translates to time noise (*i.e.* jitter) directly via the slope $\frac{dI}{dt}$ at the detected transition. That is, the timing jitter is $\sigma_\tau = \sigma_I / \left| \frac{dI}{dt} \right|$, where σ_I is the current noise. This was explained in the previous round of reviewing, and it is also written on page 4 of the supplemental material. The method can be found e.g. in Ref. [1] Section 8.3.3: Noise Creates Jitter.

[1] W. Maichen. Digital Timing Measurements: From Scopes and Probes to Timing and Jitter. Frontiers in Electronic Testing. Springer US, New York, NY, 2006.

Reviewer #1: More important, I do not see how one can get correlation data on time scales less than the filter time of the data. Assume, for simplicity, that a brick wall low pass filter is employed (although I assume that the authors use something more sophisticated). Then the Fourier transform of the cross-correlation is zero beyond a frequency of 200 Hz, which means that the frequency range over which the cross-correlation is calculated is also terminated at 200 Hz, so that the minimum time over which the discrete real time cross-correlation can be calculated is 4 ms or so. Anything less is not meaningful.

I may be wrong, but I would like the authors to show mathematically why this is incorrect.

Our response: As described in the response to the first report, the cross-correlations can be detected at shorter times than the detector risetime. This is possible since we can determine the time instance at which the event happens at a sub-risetime precision. The timing jitter (discussed above) determines the accuracy for this. Our experiment supports this observation without any assumptions: We observe coincidental events with less than 1 ms separation in time. A crucial point here is that we make no assumptions on the timing jitter or the bandwidth when analyzing the experiment. In the previous round of reviewing, we added Fig. 3 in the supplemental material together with a text that describes this important point. Here, we will refrain from going into a discussion about the example described by the Reviewer as it is not relevant for the way in which we analyze our experimental data. By contrast, it may lead to further confusion.

Reviewer #1: In the same vein, the authors now also give new information that the data are acquired using preamplifiers with a bandpass of 1 kHz before being digitized. I assume these are low pass filtered: again, I am not sure why the data are digitized at 20 kHz when the data being pre-filtered at 1 kHz?

Our response: By doing so, we obtain extra measurement points, which provide further support of our analysis.

Reviewer #1: I am also a bit confused by the discussion of the feedback loops used to keep the islands at the charge neutrality point. It seems that these are only used in between measurements to correct for drifts. Are they in operation during the measurements as well? If so, the frequency response of the feedback loops will also come into play in the analysis, unless the time scales are vastly different from the time scales of interest.

Our response: Yes, the feedback is only used between the measurements as described in the supplemental material. The Reviewer has indeed understood this correctly and therefore there is no issue during the measurement time.

Reviewer #1: Since this is a central point of the paper, I believe it needs to be resolved before the paper is suitable for publication.

Our response: We have already addressed the above points in the previous round of reviewing, and as pointed out by Reviewer #2 below, we have correctly addressed these concerns. Any remaining confusion, we believe, are due to the quantum optics approach we have taken, which is different from conventional solid-state approaches.

Response to Second Report of Reviewer #2:

Reviewer #2: The authors have revised their manuscript along the suggested lines from both the reviewers. They have in particular correctly addressed the sharp questions of reviewer #1. I recommend publication in Nature Communications.

Our response: We thank the Reviewer for the careful reading of our response and the revised manuscript, and we are happy that the Reviewer now recommends publication of our manuscript in Nature Communications. We also thank the Reviewer for confirming that we have correctly addressed all remaining concerns.

REVIEWERS' COMMENTS

Reviewer #2 (Remarks to the Author):

The authors have responded correctly to the recommendations by reviewer #1 and #2 in my view. It seems to me that there is a small confusion by reviewer #1 between time response of the detector and the timing accuracy of the charge state sequences presented in the paper. Such a trick is very often used in many time domain measurements on quantum systems (not only for charge detection like here).

It is also clear in my view from the data that noise on the detector (current noise here) results in timing accuracy. So, in my view, I do not see any further reason to delay the publication of the present manuscript as it is.